# Multidimensional Approach of Heart Failure Diagnosis and Prognostication Utilizing Cardiac Imaging with Biomarkers

**DOI:** 10.3390/diagnostics12061366

**Published:** 2022-06-01

**Authors:** In-Cheol Kim, Byung-Su Yoo

**Affiliations:** 1Division of Cardiology, Department of Internal Medicine, Keimyung University Dongsan Hospital, Keimyung University College of Medicine, Daegu 42601, Korea; kimic@dsmc.or.kr; 2Division of Cardiology, Department of Internal Medicine, Yonsei University Wonju College of Medicine, Wonju 26426, Korea

**Keywords:** heart failure, diagnosis, prognosis, multimodality imaging, biomarkers

## Abstract

Heart failure (HF) is a clinical syndrome caused by various etiologies that results in systolic and diastolic cardiac dysfunction with congestion. While evaluating HF and planning for treatment, physicians utilize various laboratory tests, including electrocardiography, diverse imaging tests, exercise testing, invasive hemodynamic evaluation, or endomyocardial biopsy. Among these, cardiac imaging modalities and biomarkers are the mainstays during HF diagnosis and treatment. Recent developments in non-invasive imaging modalities, such as echocardiography, computed tomography, magnetic resonance imaging, and nuclear imaging, have helped us understand the etiology, pathophysiology, and hemodynamics of HF, and determine treatment options and predict the outcomes. Due to the convenience of their use and potential impact on HF management, biomarkers are increasingly adopted in our clinical practice as well as research purpose. Natriuretic peptide is the most widely used biomarker for the diagnosis of HF, evaluation of treatment response, and prediction of future outcomes. Other cardiac biomarkers to evaluate the pathophysiological mechanisms of HF include myocardial injury, oxidative stress, inflammation, fibrosis, hypertrophy, and neurohormonal activation. Because HF results from complex cardiac disorders, it is essential to assess the disease status multidimensionally. The proper utilization of multimodality imaging and cardiac biomarkers can improve the quality of patient management and predict clinical outcomes in HF in the era of personalized medicine.

## 1. Introduction

Heart failure (HF) is a clinical syndrome caused by various etiologies, resulting in systolic and diastolic cardiac dysfunction with congestion [1,2,3,4,5]. Recent HF guidelines have classified HF into three categories according to the ejection fraction (EF): HF with reduced EF (HFrEF, EF ≤ 40%), HF with mildly reduced EF (HFmrEF, 40% < EF < 50%), and HF with preserved EF (HFpEF, EF ≥ 50%) [3]. The evaluation of HF requires additional information, including laboratory tests, electrocardiography, diverse imaging tests, exercise testing, invasive hemodynamic evaluation or endomyocardial biopsy [1]. Among these, multimodality imaging and cardiac biomarkers are the most frequently used methods for HF diagnosis and treatment. Recent developments in non-invasive imaging modalities can help assess the causes, pathophysiology, and hemodynamics of HF, determine treatment options, and predict outcomes. The role of biomarkers in HF has also been increasingly focused on [6,7]. Natriuretic peptide is the most important biomarker for HF diagnosis, evaluation of treatment response, and prediction of outcome [8,9]. Other cardiac biomarkers, such as myocardial injury, oxidative stress, inflammation, fibrosis, hypertrophy, and neurohormonal activation, are being introduced regularly to assess the pathophysiological mechanisms of HF [6,10,11,12]. Because HF is a caused by complex cardiac disorders, it is critical to assess heart condition multidimensionally to improve patient management [3,5,13]. In this review, we describe a multidimensional approach using various imaging modalities and multiple biomarkers in HF management for risk stratification, diagnosis, etiology evaluation, treatment, and prognostication.

## 2. Diagnosis of Heart Failure in Recent Guidelines

The recently proposed HF definition suggests that HF is a clinical syndrome with current or prior symptoms and/or signs caused by a structural and/or functional cardiac abnormality (as determined by EF < 50%, abnormal cardiac chamber enlargement, E/e′ > 15, moderate/severe ventricular hypertrophy, or moderate/severe valvular obstructive or regurgitant lesion) and corroborated by at least one of the following: elevated natriuretic peptide levels; objective evidence of cardiogenic pulmonary or systemic congestion by diagnostic modalities, such as imaging (e.g., chest X-ray or elevated filling pressures by echocardiography) or hemodynamic measurement (e.g., right heart catheterization, pulmonary artery catheter) at rest or with provocation (e.g., exercise) [1] (Figure 1). NT-proBNP or BNP tests should be performed when HF is suspected based on risk factors, symptoms and/or signs, or abnormal electrocardiography. Supposing that the result is abnormal or the laboratory test is not available, further evaluation of the echocardiography needs to be performed to confirm the diagnosis of HF and classify the phenotype of HF according to the EF [3]. Finally, the etiology should be determined to establish the treatment strategy. The utilization of various biomarkers and multimodality imaging is helpful during the whole step of HF, including diagnosis through the future prognostication of the disease.

## 3. Multimodality Imaging Approaches in Various HF

### 3.1. Multimodality Imaging in HF

The first step of imaging in HF is determining the signs of congestion by chest X-ray. According to the recent HF guidelines, the HF can be classified through transthoracic echocardiography (TTE). [1,3,4] In HF, ruling out ischemic etiology is most important because the treatment needs to be applied depending on the obstruction of the coronary arteries. Other etiologies such as idiopathic dilated cardiomyopathy (DCM), burn-out stage of hypertrophic cardiomyopathy (HCM), and restrictive cardiomyopathy (RCM) combined with systolic dysfunction should be further investigated by utilizing other invasive and non-invasive imaging modalities [14]. The appropriate choice of imaging modality is crucial in the management of HF [15] (Table 1).

### 3.2. Echocardiography

Conventional 2-dimensional TTE (2D TTE) is widely used to evaluate chamber size, systolic function, and diastolic function [3]. Regional wall motion abnormalities observed in 2D TTE suggest combined coronary artery disease or focal myocardial fibrosis caused by inflammation, such as sarcoidosis [14,16]. In patients with a poor echocardiographic window, contrast echocardiography can be performed to clearly distinguish the endocardial border. In contrast, by echocardiography with pharmacologic stress agents (e.g., dobutamine, and adenosine), we can observe regional wall motion abnormalities and myocardial perfusion defects, which indicate the presence of coronary artery obstruction, inducible ischemia, and viability according to the response to the pharmacological agent. Vortex flow analysis was recently shown to be helpful in predicting clinical outcomes as well as exercise capacity [17,18]. Volumetric information and valvular dysfunction can be assessed more easily using 3-dimensional (3D) TTE in a shorter time by recently developed vendors [15,19]. Observation of the inferior vena cava (IVC) at a subcostal window by echocardiography can guide the volumetric assessment and fluid responsiveness to various degrees of HF [20]. Hypovolemic patients can be identified by the observation of IVC. Normal collapse of IVC during deep inspiration is not observed in the case of increased right atrial pressure (RAP) [21]. Estimation of RAP is guided by the combination of the size and collapsibility of IVC. Routine measurements in the size of the IVC and collapsibility with respiration in patients with HF reliably guide fluid management decisions [22].

### 3.3. Lung Ultrasound

Lung ultrasound is an additive tool to evaluate heart failure status in a couple of minutes using general cardiac transducer by echocardiography (3.5 to 5.0 MHz) [23,24]. The main target is pleural water (pleural effusion) and lung water (pulmonary congestion as multiple B-lines). Lung ultrasound provides accurate information over simple chest radiography with a low-cost, portable, real-time, radiation-free method. The BLUE (bedside lung ultrasound in emergency) protocol has proposed a standardized approach to the most common causes of acute respiratory failure, such as pneumonia, acute pulmonary edema, chronic obstructive pulmonary disease, asthma, pulmonary embolism, and pneumothorax [25]. Adequate decongestion can be applied in accordance with the result of the lung ultrasound in acute HF.

### 3.4. Cardiac Computed Tomography

The use of cardiac computed tomography (CCT) is rapidly increasing. In addition to coronary artery evaluation, anatomical and functional evaluation of various cardiac structures are available with higher resolution when applying the retrospective protocol [15,16,19,26,27]. With the advancements in scanner design, technology, and computer processing power, CCT has emerged as a valuable method for the diagnosis, treatment and prognostication in HF. Anatomical information, including chamber quantification, coronary, valvular evaluation with functional information by myocardial perfusion imaging, fractional flow imaging, and regional ventricular function evaluation, can be the advantage of CCT. It is also useful in the field of electrophysiology, valvular disease, and end-stage heart failure after the left ventricular assist device or heart transplantation [28]. The effort to avoid semi-invasive transesophageal echocardiography by CCT is in line with the development of the techniques and protocols to decrease the radiologic dose with reduced contrast agents [29]. 

### 3.5. Nuclear Imaging

Nuclear imaging is a frequently used non-invasive functional imaging technique. Single positron emission computed tomography (SPECT) with ²⁰¹Thallium chloride or ⁹⁹mTechnetium-sestamibi can assess cell membrane integrity and mitochondrial function, as well as myocardial perfusion. Myocardial perfusion can also be assessed with ¹³N-ammonia and ⁸²Rubidium-PET [15,16,27]. Physicians can assess glucose metabolism by using positron emission tomography (PET) imaging with ¹⁸F-fluorodeoxyglucose (¹⁸F-FDG) to evaluate myocardial viability. Preserved FDG update in an area with LV systolic function with reduced myocardial perfusion indicates preserved viability [16,27].

### 3.6. Cardiac Magnetic Resonance Imaging

Cardiac magnetic resonance (CMR) is helpful for evaluating myocardial function and anatomical evaluation. Both T1 and T2 imaging sequence variations allow for the assessment of myocardial inflammation and edema as frequently seen in myocarditis or inflammatory cardiomyopathies [30,31]. The pattern of late gadolinium enhancement (LGE) on CMR is helpful in differentiating the etiology of the underlying disease [14,31]. In ischemic cardiomyopathy, subendocardial or transmural LGE corresponding to the coronary artery territories is frequently noted [16]. In non-ischemic cardiomyopathy, mid-wall LGE (idiopathic DCM, myocarditis, sarcoidosis, Anderson–Fabry disease, Chagas disease, HCM, and right ventricular pressure overload), epicardial LGE (sarcoidosis, myocarditis, Anderson–Fabry disease, and Chagas disease), and global endocardial LGE (amyloidosis, systemic sclerosis, and post-cardiac transplantation) can be found according to the entity [26,31,32].

## 4. Biomarkers in Heart Failure

With the development of novel HF medications, biomarkers have emerged as a helpful guide in the management of HF. Various biomarkers have been developed for diagnosis, management, and prognostication. The most widely adopted biomarkers for diagnosis, management, and prognostication are natriuretic peptides (B-type natriuretic peptide and N-terminal-proBNP). Many other biomarkers have been developed to evaluate pathophysiological processes related to HF, such as myocardial insult, neurohormonal activation, myocardial inflammation, and remodeling. Some of these biomarkers have already been utilized in clinical practice (troponin, sST2, NGAL, and galectin-3). However, other biomarkers are still in research or are only used to evaluate pathophysiological mechanisms of specific types of HF. Likewise, for the imaging modalities, a multidimensional strategy with different biomarkers is required in each step of HF management (Figure 2).

### 4.1. Markers for Myocardial Insult

(1)B-type natriuretic peptides

Natriuretic peptides are synthesized by the brain, heart, and other organs. In the heart, these hormones are provoked by atrial and ventricular distention and neurohormonal stimulations in response to HF. BNP is originally recognized in the brain but is discharged predominantly from the ventricles of the heart in reaction to volume expansion and overload [11]. ProBNP is produced from the prepro-peptide, and it is cleaved into the active form of BNP and the biologically inert form of NT-proBNP. The biological effects of BNP include diuresis, natriuresis, vasodilation, improved lusitropy, and reduced cardiovascular hypertrophy and fibrosis. Several enzymes, including neprilysin, degrade BNP. However, NT-proBNP is passively cleared by numerous organs, including the kidneys. BNP or NT-proBNP concentrations increased significantly in HF with myocardial stretch. In addition, cardiac functional and structural abnormalities and combined conditions may trigger the elevation of both BNP and NT-proBNP levels. There is concrete evidence that the concentration of BNP or NT-proBNP is closely related to the certainty of HF diagnosis, and the severity of HF [33,34]. When the interpretation of BNP and NT-proBNP was made with clinical decision making, the diagnostic ability increased, and outcomes improved with only small additive cost [33,35,36,37].

The negative predictive value of both BNP and NT-proBNP is 96% (cut-off value < 30–50 pg/mL) and 99% (cut-off value < 300 pg/mL), respectively, to exclude acute decompensated HF. In patients with an outpatient clinic, the negative predictive value remained high (96% for BNP with cut-off value < 20–40 pg/mL, 98% for NT-proBNP with cut-off value < 125 pg/mL) [7]. Physicians would exclude HF when the level is normal in the clinical practice, especially when the symptom is atypical. According to the previous studies and guidelines, BNP normal ranges were <35 pg/mL (100 pg/mL for hospitalized patients), and NT-proBNP normal ranges were <125 pg/mL (300 pg/mL for hospitalized patients) [38,39].

Based on recent guidelines, angiotensin receptor/neprilysin inhibitor (ARNI) has become a mainstay of HFrEF treatment [38,39]. In addition to angiotensin blockade, it enhances natriuretic peptides by inhibiting neprilysin. In the PARADIGM HF trial, ARNI reduced primary endpoint (a composite of cardiovascular death or first hospitalization for HF) by 20% compared to enalapril [9,40]. When evaluating the effect of ARNI, NT-proBNP was preferred to avoid mechanistic interaction. In PIONEER-HF and PROVD-HF, a rapid and significant decrease in NT-proBNP level was noted after the initiation of ARNI [41,42]. The prognostication ability of both BNP and NT-proBNP has been well documented from the previous studies of HF [43,44,45]. 

(2)Atrial natriuretic peptide

Similar to B-type natriuretic peptides, atrial natriuretic peptide (ANP) also increases in response to myocardial wall stress. It is the first hormone isolated from the heart as a potent natriuretic/diuretic and hypotensive factor, which induces profound natriuresis/diuresis, hypotension, and inhibition of aldosterone secretion [10]. Since the half-life of ANP is short (2–5 min), proANP, which has a longer half-life, can be measured. The mid-regional propeptide assay for the ANP (MR-proANP) assay would provide adequate diagnostic power for HF. MR-proANP was compared with previous B-type natriuretic peptides. MR-proANP ≥ 120 pmol/L was diagnostic for acute decompensated HF and was not superior to BNP or NT-proBNP [46]. In addition, MRproANP added independent prognostic information beyond NT-proBNP and relevant clinical characteristics in a reclassification analysis. In other studies, MR-proANP revealed non-inferiority for the diagnosis of HF compared to NT-proBNP. Furthermore, the level of MR-proANP was also helpful in predicting poor outcomes [47,48]. In the GISSI-HF study, the MR-proANP ≥ 278 pmol/L showed fair prognostic accuracy for 4-year mortality in chronic HF (AUC = 0.74; 95% CI, 0.70–0.76) [48].

(3)High-sensitivity troponin

High-sensitivity troponin (hsTn) is a representative biomarker for myocardial injury in patients with myocardial infarction or acute myocarditis. Elevated serum concentrations of hsTn are also frequently observed in patients with various stages of HF despite the absence of coronary artery obstruction. The release of cardiomyocyte cell membrane blebs during myocardial wall stress may be the reason for elevated troponin levels in HF. Elevated serum concentrations of hsTn are associated with adverse clinical outcomes and cardiac remodeling in HF [49,50,51]. In a study of patients with LV systolic dysfunction, higher hsTn level was associated with a higher risk of clinical events (*p* = 0.008), and more time in low hsTn (≤10.9 pg/mL) was also associated with a lower cardiovascular event rate after adjustment (OR, 0.81; *p* = 0.008) [52].

(4)Myeloperoxidase

Myeloperoxidase (MPO) is a biomarker released by activated neutrophils and monocytes in response to oxidative stress during inflammation. The depletion of vascular nitric oxide and promotion of low-density lipoprotein oxidation in endothelial cells are the major contributors. Increased serum MPO concentration is associated with poor outcomes in HF, and it is more pronounced when combined with BNP [53,54]. Increased MPO concentrations (>99 pmol/L) were associated with significantly higher one-year mortality in patients with acute decompensated HF (HR, 1.58; *p* = 0.02) [53]. Higher MPO levels are also associated with higher BNP levels and symptoms (*p* < 0.0001) [54].

(5)Uric acid

Uric acid is a marker of cell stress that predicts HF onset. A high uric acid level predicts a higher incidence of HF and an increased risk of adverse events [55]. A previous study revealed that the decrease in uric acid by xanthine oxidase improved endothelial function in HF; however, it was not associated with the improvement in the composite events, including hospitalization and cardiovascular death [55].

### 4.2. Markers for Neurohormonal Activation

In HF, activation of the sympathetic nervous system results in elevated norepinephrine concentration, which is responsible for increased heart rate, myocardial contraction, peripheral vascular resistance, and energy expenditure. Increased plasma concentrations of norepinephrine are associated with increased cardiovascular risk [56]. In advanced HF, the myocardial concentration of norepinephrine paradoxically decreases probably due to the depletion of the receptor at the end-stage of the disease [57].

(1)Adrenomedullin

Although the fundamental mechanism is not fully understood, the endogenous vasodilator adrenomedullin (ADM) helps in the diagnosis of HF. Mid-regional propeptide (MR-proADM), which is a precursor of ADM, can be measured due to the short half-life of ADM. High MR-proADM is related to HF diagnosis and is helpful in diagnosis, independent of the NT-proBNP levels. Furthermore, the MR-proADM level is an independent predictor of death [47]. Another study showed that MR-proADM is a predictor of poor survival at 12 months (HR, 1.82; 95% CI, 1.24–2.66; *p* = 0.002), similar to NT-proBNP [58].

(2)Arginine vasopressin

The antidiuretic and vasoactive hormone arginine vasopressin (AVP) is released from the hypothalamus in response to changes in plasma osmolality and blood volume [10]. In severe HR, AVP levels are elevated in patients with severe HF. The C-terminal segment of provasopressin was measured instead of an unstable AVP. The level of the C-terminal segment of provasopressin is associated with adverse outcomes in patients with acute HF independent of NT-proBNP and troponin [12,59]. AVP utilization can be considered in patients who might benefit from vasopressin receptor antagonists, but more evidence needs to be accumulated in the future.

(3)Endothelin-1

In response to inflammation, neurohormonal activation, and vascular shear stress, endothelin-1 (ET-1) is released by the vascular endothelium [10]. It is responsible for vasoconstriction, inflammatory reactions, proliferative action, and free radical formation. High levels of ET-1 are associated with LV diastolic dysfunction and poor clinical outcome [60].

### 4.3. Markers for Inflammation and Myocardial Remodeling

(1)Tumor necrosis factor, interleukin-6

Inflammatory biomarkers, such as tumor necrosis factor (TNF), soluble TNF receptors, and interleukin-6 (IL-6) are also related with the HF outcome. In chronic HF, inflammatory markers are related to increased mortality [61,62] Microbiomes are thought to be associated with HF outcome in terms of inflammation in the right HF.

(2)Soluble suppression of tumorigenicity 2

Under myocardial or vascular strain conditions, a member of the IL-1 receptor family, ST2, is released. Among the two types of ST2 isoforms (ST2 ligand and soluble ST2), soluble ST2 (sST2) has a more critical role in myocardial and vascular remodeling, fibrosis, and atherosclerosis [63]. The concentration of sST2 can predict mortality in both acute and chronic HF [64,65]. In the PRIDE study, elevated sST2 predicted death at one year in chronic HF (HR, 5.6; 95% CI, 2.2–14.2; *p* < 0.001) and acute HF (HR, 9.3; 95% CI, 1.3–17.8; *p* = 0.03) above and beyond NT-proBNP [66]. When combined with NT-proBNP, the accuracy of prediction for poor outcomes increased [66]. The role of sST2 in risk prediction persisted in both HFpEF and HFrEF [67]. The benefit of sST2 over natriuretic peptides is that it is not affected by patient conditions, such as obesity, age, atrial fibrillation, or renal function. Previous studies have shown that serial measurements of sST2 predict outcomes better than single measurements in various types of HF [68]. The change in sST2 levels is also helpful for evaluating the HF treatment response [69].

(3)Galectin-3

Galectin-3 is a macrophage lectin product that is associated with tissue fibrosis. In HF, galectin-3 plays an essential role in LV remodeling. The concentration of galectin-3 is linked to outcomes such as death and HF hospitalization, and when it is combined with NT-proBNP, the risk prediction accuracy improves [70]. However, galectin-3 is also affected by other organs, such as the kidney, which might diminish its ability in HF [12]. The interaction between galectin-3 and aldosterone-mediated fibrosis is poorly understood, but aldosterone antagonists seem more effective in HF patients with low galectin levels. 

(4)Matrix metalloproteinases

During the development and progression of HF, cardiac extracellular matrix remodeling is a critical pathway through degradation of collagen and other matrix proteins by collagenases, matrix metalloproteinases (MMP), and by tissue inhibitors of metalloproteinases (TIMP) [10]. An imbalance between MMPs and TIMPs may lead to fibrosis and ventricular remodeling [71]. They are also known as markers of cardiac extracellular matrix remodeling during the progression of HF [10]. Elevated MMPs and TIMPs are related to cardiac fibrosis and ventricular remodeling which result in HF hospitalization in hypertensive HF patients [71]. Previous studies have shown that MMP-2, MMP-3, MMP-8, MMP-9, and TIMP-1 concentrations are associated with poor outcomes in HF patients [72,73,74,75,76].

(5)Growth/differentiation factor-15

Growth differentiation factor-15 (GDF-15) is also known as macrophage inhibitory cytokine-1 (MIC-1) due to its inhibitory function of macrophage activation. It is a part of the transforming growth factor beta (TGF-β) cytokine superfamily [77]. It regulates cellular mitochondrial function that is related with inflammation, oxidative stress, apoptosis, immune reaction, fibrosis, reparation, and malignancies [78,79]. The concentration of GDF-15 is upregulated in response to severe injury to the heart, kidney, liver, and the lung [80,81]. Although GDF-15 has been noted for its anti-apoptotic, anti-inflammatory, and antihypertrophic actions, it also noted that elevated values of GDF-15 were found to correlate with a number of cardiac diseases, including LV hypertrophy, stable coronary artery disease, myocardial infarction, and HF [11,77]. Elevated concentrations of GDF-15 predicted a greater risk of death and HF hospitalization in patients with acute and chronic HF [82,83,84]. In Val-HeFT study, the baseline GDF-15 level was associated with higher mortality (HR, 1.02; 95% CI, 1.014–1.019; *p* < 0.001) and first morbid event (HR, 1.02; 95% CI, 1.017–1.023; *p* < 0.001) [84].

## 5. Multidimensional Approach in HF with Cardiac Imaging and Biomarkers

The use of diverse imaging modalities and biomarkers for the management of HF has evolved tremendously. Echocardiography and natriuretic peptides are the most effective tool throughout the HF. However, the appropriate use of different imaging modalities and biomarkers is key to successful treatment (Figure 3). Physicians should be aware of the benefits and limitations of each test that needs to be performed in every circumstance. Many new imaging techniques and biomarkers are currently under investigation that can be adopted to improve management quality, resulting in better outcomes. A multidimensional approach in HF with cardiac imaging with biomarkers in our routine clinical practice could yield optimal results by the synergistic role.

## Figures and Tables

**Figure 1 diagnostics-12-01366-f001:**
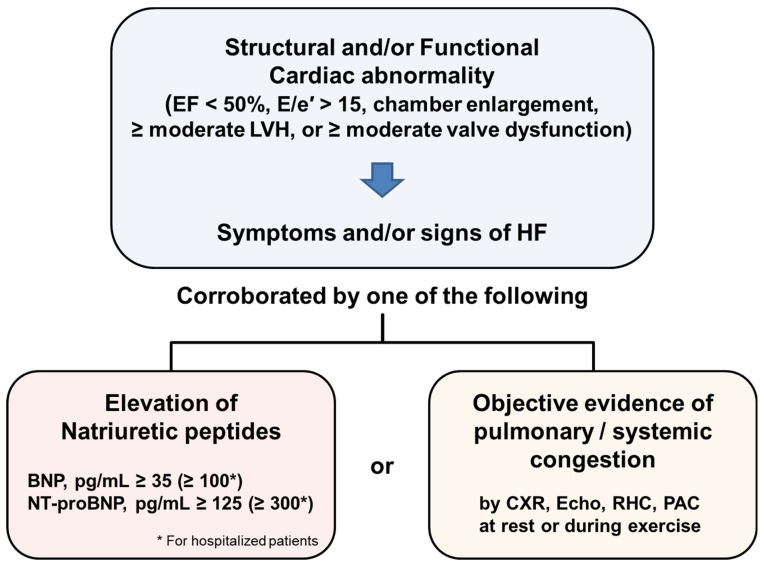
Universal definition of heart failure according to the recent guideline. EF, ejection fraction; LVH, left ventricular hypertrophy; HF, heart failure; BNP, brain natriuretic peptide; CXR, chest X-ray; echo, echocardiography; RHC, right heart catheterization; PAC, pulmonary artery catheter.

**Figure 2 diagnostics-12-01366-f002:**
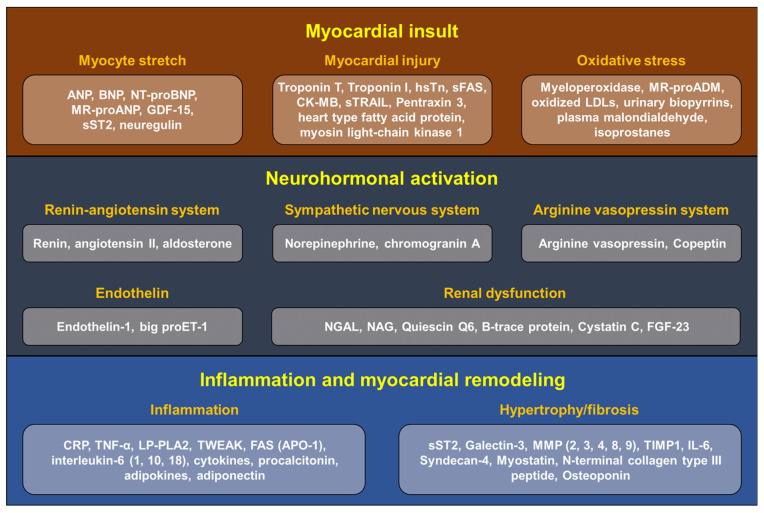
Biomarkers in heart failure according to the pathophysiological mechanism. ANP, atrial natriuretic peptide; BNP, B-type natriuretic peptide; NT-proBNP, N-terminal pro-B-type natriuretic peptide; MR-proANP, midregional pro-atrial natriuretic peptide; GDF-15, growth differentiation factor-15; sST2, soluble ST2; hsTn, high-sensitivity troponin; sFAS, soluble Fas cell surface death receptor; CK-MB, creatinine kinase-muscle/brain; sTRAIL, soluble tumor necrosis factor-related apoptosis-inducing ligand; MR-proADM, midregional proadrenomedullin; LDL, low density lipoprotein; proET-1, proendothelin-1; NGAL, neutrophil gelatinase-associated lipocalin; NAG, N-acetyl-β-D-glucosaminidase; FGF-23, fibroblast growth factor-23; CRP, C-reactive protein; TNF-α, tumor necrosis factor-α; LP-PLA2, lipoprotein-associated phospholipase A2; TWEAK, TNF-like weak inducer of apoptosis; FAS, Fas cell surface death receptor; APO-1, apolipoprotein-1; MMP, matrix metalloproteinases; TIMP1, tissue inhibitors of metalloproteinases1;IL-6, interleukin-6.

**Figure 3 diagnostics-12-01366-f003:**
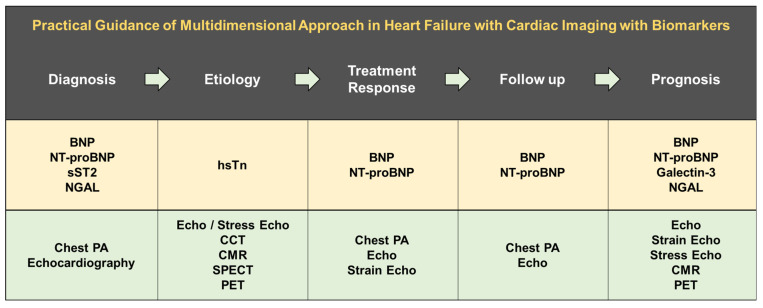
Practical guidance of multidimensional approach in heart failure with cardiac imaging with biomarkers. BNP, B-type natriuretic peptide; NT-proBNP, N-terminal pro-B-type natriuretic peptide; sST2, soluble ST2; NGAL, neutrophil gelatinase-associated lipocalin; hsTn, high-sensitivity troponin; Echo, echocardiography; CCT, cardiac computed tomography; CMR, cardiac magnetic resonance; SPECT, single-photon emission computed tomography; PET, positron emission tomography.

**Table 1 diagnostics-12-01366-t001:** The role of each multimodality imaging for the evaluation of various conditions in HF.

	Echo	Stress Echo	CCT	CMR	SPECT	PET
**Chamber size**	**+++**	**+++**	**+++**	**+++**	**-**	**-**
**Systolic function**	**+++**	**+++**	**+++**	**+++**	**++**	**-**
**Diastolic function**	**+++**	**+++**	**-**	**-**	**-**	**-**
**Valve function**	**+++**	**+++**	**++**	**++**	**-**	**-**
**Intracardiac pressure**	**+++**	**+++**	**-**	**-**	**-**	**-**
**Coronary artery disease**	**+**	**++**	**+++**	**++**	**+++**	**++**
**Myocardial viability**	**-**	**+++**	**-**	**+++**	**++**	**+++**

Echo, echocardiography; CCT, cardiac computed tomography; CMR, cardiac magnetic resonance; SPECT, single-photon emission computed tomography; PET, positron emission tomography. The number of + indicates the grade of usefulness of each imaging modality. “-“ indicates no benefit for the evaluation of cardiac condition in HF.

## Data Availability

Not applicable.

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
