# Peer review of "Multidimensional Approach of Heart Failure Diagnosis and Prognostication Utilizing Cardiac Imaging with Biomarkers"

_diagnostics, 2022, doi:10.3390/diagnostics12061366_

Round 1
Reviewer 1 Report
- overall is a well-written review which summarizes the current knowledge regrading imaging and biomarkers on HF topic.However, since the aim of the authors is to write a review article, the manuscript would be benefit from a more extensive list of referneces on each topic included.
- In multimodality imaging paragraph, as multidimensional approach is highlighted, data regarding lung sonography and inferior vena cava (IVC) estimation should be included. The combined imaging of echocardiography, lung ultrasound and IVC evaluation are included in HF diagnostic imaging modalities, according to the recent bibliorgaphy and guidelines.
- In paragraph "markers of myocardial insult" there is lack of cherence in terms of writing flow
- All figures are missing in the dowloaded material
- The language could be more formal in some parts of the manuscript. Examples: The importance of various biomarkers in HF is increasingly focused on because of its convenience and potential impact AND the conclusion paragraph. Rephrasing would improve it.
Reviewer 2 Report
In-Cheol Kim et al. made a narrative review of cardiac biomarkers.
HF can be the terminal condition of many cardiovascular diseases, including myocardial infarction (MI), valvular heart disease, and various cardiomyopathies. HF has multiple causes, and is seldomly addressed by the authors in which the biomarkers would be of advantage. Although the presenting symptoms and the syndrome of HF are similar to those arising from other causes, the three subtypes of HF also have some noteworthy features and thus differ between imagiologic and serologic biomarkees.
Due to these unique characteristics, various roads of basic and translational investigations and treatments have been developed, first to enlighten pathophysiologic routes and also to predict and prevent hospitalization and death. The authors have shed some light on this matter with this narrative review of the major and emerging markers.
The authors should further explain the pathopshiologic routes intrinsic and between some of the biomarkers ANP, NT pro BNP, GDF 15, MMP
The authors failed to quantify in the major biomarkers the relationship between them and outcomes –Heart failure risk or long term outcomes.
ex: aOR: --- (CI 95%: xx-xx) and a long term risk of aHR ---
Avoid sentences like “The negative predictive value of both BNT and NT-proBNP is excellent,”. Please quantify
The paragraph regarding Cardiac Computed tomography needs more information.
Reviewer 3 Report
I read the manuscript sent by you carefully and I found it interesting and well documented. Unfortunately, your manuscript does not contain the figures you are referring to, so I cannot appreciate the manuscript at its true value. Please complete the manuscript with figures 1, 2, 3 as these are not available in the submitted manuscript or in the supplementary materials section.
Round 2
Reviewer 1 Report
the manuscript has been sufficiently improved and all reviewer's comments had answered adequately I agree with publication in the revised formReviewer 3 Report
I have analyzed the manuscript corrected according to the requirements of the reviewers and I can say that the current form is much closer to reality and much more complete than the initial version.
You have made a very useful review of the biomarkers currently used for the diagnosis and prognosis of heart failure and I believe that the article can be published in its current form.